# Privately Aligning Language Models with Reinforcement Learning

**Fan Wu**[1,*]**, Huseyin A. Inan**[2]**, Arturs Backurs**[3]**,**
**Varun Chandrasekaran**[1]**, Janardhan Kulkarni**[3]**, Robert Sim**[2]
[1] University of Illinois Urbana-Champaign, [2] M365 Research, [3] Microsoft Research
`{fanw6,varunc}@illinois.edu`,
`{huseyin.inan,arturs.backurs,jakul,rsim}@microsoft.com`

## Abstract

Positioned between pre-training and user deployment, aligning large language models (LLMs) through reinforcement learning (RL) has emerged as a prevailing strategy for training instruction following-models such as ChatGPT. In this work, we initiate the study of privacy-preserving alignment of LLMs through Differential Privacy (DP) in conjunction with RL. Following the influential work of Ziegler et al. (2020), we study two dominant paradigms: (i) alignment via RL without human in the loop (e.g., positive review generation) and (ii) alignment via RL from human feedback (RLHF) (e.g., summarization in a human-preferred way). We give a new DP framework to achieve alignment via RL, and prove its correctness. Our experimental results validate the effectiveness of our approach, offering competitive utility while ensuring strong privacy protections.

## 1 Introduction

Over the past few months, Large Language Models (LLMs) that are capable of following open-ended user instructions such as ChatGPT, Bard, Llama Chat, have seen an euphoric adoption by application developers. Similar to their predecessors, these models are pre-trained on vast amounts of public internet data. However, their magical ability to follow myriad user instructions – the driving force behind their mass adoption – has been attributed to *instruction fine-tuning and learning from human feedback*. This new step involves collecting a dataset of human preferences and feedback, followed by fine-tuning the model via reinforcement learning (RL) to make them better *aligned*, often abbreviated as RLHF. Since the influential works of Ziegler et al. (2020); Ouyang et al. (2022); Bai et al. (2022), the RLHF framework has emerged as the dominant paradigm for training instruction-following models.

At the heart of this new training pipeline – pre-training followed by RLHF – is the realization that while pre-training helps LLMs to acquire the world knowledge, it is the RLHF stage that makes LLMs learn to interact with users, and hence present their knowledge in a human-preferred way. This framework opens the door for a continuous improvement of the model by collecting users' feedback and preferences via telemetry data. As appealing as that may sound, improving LLMs via users' preferences and feedback raises privacy concerns: what if the model learns about a specific user's instructions and regurgitates them at a later point? It is well known in the privacy literature that LLMs are vulnerable to privacy attacks including prompt attacks (Duan et al., 2023; Carlini et al., 2021; 2019), and RLHF training seems particularly concerning from this angle. This constitutes the central question explored in this work.

*Can we fulfill the promise of aligning models with human preferences and feedback data via a privacy preserving RLHF methodology?*

### 1.1 Our Contributions

We initiate the study of aligning LLMs with RL while satisfying the strong mathematical guarantees of differential privacy (DP) (Dwork & Roth, 2014). We foresee this as an important research direc-

---

*This work was carried out as part of an internship at Microsoft Research.

tion for the privacy community as more applications start to deploy LLMs to interact directly with users. Our main contributions are:

1. We give a differentially private framework for aligning LLMs with RL. Our framework mathematically guarantees that the final model satisfies DP over the entire course of the alignment process, consisting of multiple stages of model training and weight sharing. Further, we show how to adapt the PPO algorithm (Schulman et al., 2017) to DP setting.

2. We empirically evaluate our DP framework on commonly studied tasks (in non-privacy literature). Following the influential work of Ziegler et al. (2020), we evaluate two main scenarios: (i) alignment via RL without human in the loop for a positive review generation task on the IMDb dataset (Maas et al., 2011), and (ii) alignment via RL from human feedback (RLHF) for a summarization task on the Reddit TL;DR dataset (Völske et al., 2017). *Our experimental results indicate that privately aligning LLMs is possible, offering competitive utility while ensuring strong privacy protections.* As a representative example, on the IMDb dataset, the average reward obtained by our DP GPT2-Large model for generating positive reviews is 3.20 with $\epsilon = 4$, whereas the *best* performing non-private model achieves an average reward of 3.45.

Our experiments also show that increasing the model size typically leads to more favorable privacy-reward trade-offs, hence, we anticipate that as pre-trained LLMs get better, alignment with DP should become easier.

## 2 PRELIMINARIES

### 2.1 ALIGNING LANGUAGE MODELS VIA REINFORCEMENT LEARNING

We review the pipeline from the seminal work by Ziegler et al. (2020), which describes a methodology to align language models via RL by using a reward model to optimize for a given task. For ease of presentation, we borrow terminology and notations from Ziegler et al. (2020).

One starts with a pre-trained language model $LM^{\text{pt}}$, which defines a probability distribution $LM^{\text{pt}}(x_n \mid x_1, \cdots, x_{n-1}) \in [0, 1]$ over the space of tokens $x_n \in \mathcal{V}$ given a context consisting of tokens $x_i \in \mathcal{V}$ for $i = 1, \ldots, n-1$. $\mathcal{V}$ is referred as the vocabulary of $LM^{\text{pt}}$. The first step in general is to fine-tune this model with regular supervised learning procedure (SFT). This step can be performed for various reasons such as to teach the language model a desired output behaviour (Ouyang et al., 2022) or it could be simply to train for a downstream task such as summarization (Stiennon et al., 2022). We denote the resulting model as $LM^{\text{sft}}$.

In the alignment step, a policy $\pi$, initialized as $\pi = LM^{\text{sft}}$, is further fine-tuned using RL for the underlying task. We consider two scenarios depending on whether the task is directly defined by a reward function or it is based on human judgments. We compare the two scenarios in Appendix G.

**Reinforcement learning without human in the loop.** The underlying task is defined by a reward function $r : \mathcal{V}^\infty \times \mathcal{V}^\infty \to \mathbb{R}$ that can score how well aligned the language model's generation $y \in \mathcal{V}^\infty$ is, given the context $x \in \mathcal{V}^\infty$. Here, one can use reinforcement learning to directly optimize the expected reward. An example is controlled sentiment generation, where the goal is to respond to a user query with a positive sentiment. Here, one can use existing language models that are fine-tuned on sentiment classification tasks as the reward model to score for positive sentiment.

**Reinforcement learning with human preferences.** The underlying task is defined by human judgments. A typical example is to respond to a user query with a human-preferred way instead of language model's original completion that is learned during pre-training. Here, human labels are used first to train a reward model. A dataset can be formed by generating multiple responses from the LLM (for simplicity we consider two: $y_1$ and $y_2$) for a given input $x$ and asking humans to prefer between $y_1$ and $y_2$. Let $b \in \{0, 1\}$ denote the human preference. Assuming access to a dataset $\mathcal{S}$ of $(x, y_0, y_1, b)$ samples with human preferences, a reward model $r : \mathcal{V}^\infty \times \mathcal{V}^\infty \to \mathbb{R}$ can be trained with the following negative log-likelihood loss (Ziegler et al., 2020; Ouyang et al., 2022):

$$\mathcal{L}(r, \mathcal{S}) = -\mathbb{E}_{(x, y_0, y_1, b) \sim \mathcal{S}} \left[ \log \left( \sigma \left( r(x, y_b) - r(x, y_{1-b}) \right) \right) \right], \tag{1}$$

where $\sigma$ denotes the sigmoid function: $\sigma(x) = \frac{1}{1+e^{-x}}$. One can also initialize the reward model $r$ from $LM^{\text{sft}}$ with an additional linear layer that produces a single scalar for the reward value.

Finally, the initialized policy $\pi$ is fine-tuned to optimize the reward model $r$ with reinforcement learning. However, instead of directly optimizing the expected reward, a penalty term of $\beta \text{KL}(\pi, LM^{\text{sft}})$ is added to the optimization term to prevent $\pi$ from deviating too far from $LM^{\text{sft}}$. Thus, the modified reward becomes

$$R(x, y) = r(x, y) - \beta \log \frac{\pi(y|x)}{LM^{\text{sft}}(y|x)}. \tag{2}$$

This reward function is maximized via Proximal Policy Optimization (PPO) (Schulman et al., 2017) to fine-tune the policy $\pi$ on the corresponding data distribution $x \sim \mathcal{D}$.

## 2.2 Differential Privacy

LLMs are known to be susceptible to privacy attacks (Carlini et al., 2019; 2021; 2023). Over the past decade, Differential Privacy (DP) (Dwork et al., 2006) has emerged as a powerful framework that provides mathematical guarantees for the privacy of individuals in training datasets. It quantifies the amount of information one could learn from the output of an algorithm or its generations. Formally,

**Definition 1** (($\epsilon, \delta$)-DP (Dwork & Roth, 2014)). *A randomized algorithm $\mathcal{M}$ achieves ($\epsilon, \delta$)-DP, if for any neighboring datasets $D_1$ and $D_2$ (differing in at most one entry) and for any $S \in Range(\mathcal{M})$,*

$$\Pr(\mathcal{M}(D_1) \in S) \leq e^{\epsilon} \Pr(\mathcal{M}(D_2) \in S) + \delta. \tag{3}$$

Here, $\epsilon$ represents the privacy budget: a smaller $\epsilon$ offers a stronger privacy guarantee. $\delta$ accounts for the probability that $\mathcal{M}$ violates $\epsilon$-DP.

**DPSGD.** In the context of deep learning, DPSGD (Song et al., 2013; Abadi et al., 2016), a drop-in replacement of the vanilla stochastic gradient descent, has become the default optimizer to achieve DP. At each iteration, DPSGD performs per-sample gradient clipping and Gaussian noise addition, thus limiting and masking the contribution of any single data point to the model update; we give a detailed description in Appendix A. Recently, in response to the rising concerns of privacy leakage in large language models (Carlini et al., 2021), DPSGD is employed for tasks from private fine-tuning (Yu et al., 2022; Li et al., 2022) to synthetic text generation (Yue et al., 2023).

## 3 Problem Definition

Consider the generic problem of aligning a language model towards an objective by using a reward model via RL. Our problem formulation introduces an extra dimension to this challenge, namely, achieving this alignment respecting DP of the underlying data samples. We are given privacy parameters $\epsilon > 0$, $\delta \in [0, 1]$ and a pre-trained language model $LM^{\text{pt}}$. There is a private dataset $D$ for use in the alignment procedures outlined in Section 2.1. The particular use of the private dataset $D$ depends on the availability of the reward model.

In RL without human in the loop scenario, we assume the availability of a reward model that is independent of the private dataset $D$. In this case, supervised learning step (SFT) fine-tunes $LM^{\text{pt}}$ with $D$ to achieve the initial policy $\pi = LM^{\text{sft}}$. This is directly followed by the Proximal Policy Optimization (PPO) that fine-tunes the policy $\pi$ with the reward model on $D$. The privacy-preserving constraint in our problem definition is that the final parameters of the optimized policy $\pi$ should be ($\epsilon, \delta$)-DP with respect to private dataset $D$.

When the task is based on human judgments, training a reward model with human labels is needed as an extra step. As described in Section 2.1, the reward model is obtained by training on a dataset where each sample is a tuple consisting of a sample $x$ belonging to $D$, multiple generations of $LM^{\text{sft}}$ given $x$ as context and human preference over these generations. The privacy-preserving constraint similarly follows as the previous scenario and the final parameters of the optimized policy $\pi$ must be ($\epsilon, \delta$)-DP with respect to private dataset $D$. As we will discuss shortly, this will require the reward model to be trained with DP as well.

## 4 Our DP Alignment Framework

In this section, we describe our DP framework for aligning LLMs. We consider the scenario with human in the loop as it subsumes the case without. Recall that it involves three main steps: 1)

Supervised fine-tuning of a language model for the task at hand to obtain $LM^{\text{sft}}$. 2) Learning a reward model $r$ from human preferences. 3) Fine-tune a policy $\pi$ (initialized to $LM^{\text{sft}}$) to optimize the reward model $r$ with RL. Although we only require that the weights of the final policy $\pi$ are DP with respect to $D$, one needs to perform each step with DP, the reason for which will become clear once we describe our framework.

To achieve $(\epsilon, \delta)$-DP with respect to a private dataset $D$ at the end of the alignment there is more than one solution. One could *partition* $D$ into three disjoint subsets $D_1$, $D_2$ and $D_3$ corresponding to the three stages of the alignment pipeline, and assume that two neighboring databases differ by a single sample in *one of these* three datasets; that is, a single user can contribute to at most one of three datasets $D_1$, $D_2$ and $D_3$. Another option is to assume that a single user can contribute to *all* the three datasets $D_1$, $D_2$ and $D_3$. Our framework can handle both the settings, with minor differences in how to calculate the final privacy parameters. The former approach would mean that to calculate the final privacy parameters, we can use the *parallel* composition theorem of DP (McSherry, 2009). For the latter, one needs to use advanced composition theorems such as (Gopi et al., 2021). An additional hyperparameter related to DP in the second approach is on how to allocate the fixed privacy budget across the three steps. The goal of this work is to show that alignment with DP is possible, and, hence, we take the simpler approach and assume that a single user can contribute to at most one of three datasets $D_1$, $D_2$ and $D_3$. We clarify that the nature of the three datasets are different, with $D_1$ being a labeled dataset consisting of a reference answer per sample, $D_2$ a preference dataset consisting of two generations and a human preference bit per sample, and $D_3$ an unlabeled dataset consisting of input samples only.

With this discussion behind us, we write down our framework for DP Alignment.

1. **DP Supervised Fine-Tuning:** We do a supervised fine-tuning of $LM^{\text{pt}}$ using DPSGD with privacy parameters $(\epsilon, \delta)$ on the dataset $D_1$ to obtain $LM^{\text{sft}}$. The analysis of DPSGD (Abadi et al. (2016)) guarantees that the *weights* of $LM^{\text{sft}}$ are private, and hence $LM^{\text{sft}}$ can be used arbitrarily in the remaining pipeline.
2. **DP Learning of Reward Model:** We initialize a reward model $r$ from $LM^{\text{sft}}$ with the addition of a linear layer that produces a single scalar prediction for the reward value. We train $r$ using DPSGD with the privacy parameters $(\epsilon, \delta)$ to optimize the reward objective given by Equation 1 on the dataset $D_2$.
3. **Alignment with DPPPO:** Finally, we train a policy $\pi$ initialized to $LM^{\text{sft}}$ via a DP adaptation of Proximal Policy Optimization (PPO) with the privacy parameters $(\epsilon, \delta)$ to optimize the reward $R$ as given in Equation 2 on the dataset $D_3$.

All our model training procedures use LoRA (Hu et al., 2022). While this is not standard in the alignment literature, we make this algorithmic choice due to 3 reasons: 1) DP training works better with LoRA as hyperparameters are more stable (Yu et al., 2022); 2) LoRA is computationally more efficient, especially for DP training; 3) We also conjecture that LoRA fine-tuning during RL stage can also help in ensuring that the aligned model does not drift too far away from the $LM^{\text{sft}}$ model. This may be an interesting point even in the non-private world.

We will elaborate each of the steps in our framework below. Before that, we note the following privacy guarantee of our framework due to the *parallel* composition theorem of DP (McSherry, 2009).

**Theorem 2.** *Our DP alignment framework is $(\epsilon, \delta)$-differentially private.*

**Why Step II needs to be DP?** A curious reader may ask why one should learn the reward model using DP if the third step already satisfies DP; that is, can learning the reward model be non-private? This is a subtle but important point, and our algorithmic choice of making the second step DP has to do with the privacy analysis of DPSGD. Consider the scenario where the reward model is not private. In such a case, for every random mini-batch in the third step, the gradients are a function of the reward model, which in turn implies that the gradients of the mini-batch are a function of entire dataset $D_2$. This invalidates the privacy amplification by subsampling theorem of DPSGD (Abadi et al., 2016), which is crucial for the overall framework to work.

Our solution to the above technical challenge is to learn the reward model also via DP. The post-processing theorem of DP (Dwork & Roth, 2014) guarantees that the reward model $r$ can be used in the third step as if it were a public model. However, there could be other ways of achieving the alignment with DP where the second step is not private, and we leave it as an intriguing future research question.

**DP Adaptation of PPO.** Next, we discuss our algorithmic choices in making PPO algorithm DP. Although the model updates in our DPPPO algorithm are done via DPSGD, there are some important technical details to consider. An important distinction between DPPPO and SFT with DPSGD is what governs the number of iterations and how the model weights are updated. For SFT using DPSGD (1st step), the model weights are updated for each batch, and the number of iterations governs the course of training and the total model weight updates. On the other hand, PPO performs model updates in minibatches within a batch and multiple rounds (a.k.a. PPO epochs) can be taken over the *same* batch. Further, regular epochs are taken over the full dataset; see Schulman et al. (2017) (Algorithm 1) for a precise description. We give a complete pseudo-code of PPO implementation in Appendix F, but for the sake of discussion consider the abbreviated version in Algorithm 1. PPO updates are given in lines 6-10 that needs to be privatized. In our DPPPO implementation, we set $T_{\text{PPO}} = 1$ deviating from the usual implementations in RL that set $T_{\text{PPO}} > 1$; for example, von Werra et al. (2022) defaults $T_{\text{PPO}}$ to 4. By appropriately selecting the batch size (we use larger batch size for DPPPO), we ensure that the total number of model updates in both the private and non-private worlds remain similar. We make these algorithmic choices to simplify the privacy analysis and to utilize privacy amplification by subsampling (Abadi et al., 2016) in DPSGD algorithm, where each batch should be randomly selected from the dataset. If one takes more than 1 round of model updates ($T_{\text{PPO}} > 1$) on the same batch, then privacy accounting of DPSGD needs to be modified, say by first doing an advanced composition across multiple PPO rounds on the same batch followed by subsampling amplification. We leave these algorithmic explorations on our choices as future research directions, and present an ablation study on $T_{\text{PPO}} > 1$ in Appendix B.1.

---

**Algorithm 1:** Aligning language models with RL (PPO), full version in Appendix F.

---

**Define:** $D$: a dataset consisting of input texts. $x$: input text, $y$: model response.
 $T$: total training epochs, $T_{\text{PPO}}$: PPO epochs.
 model, ref_model: the model being learned and the frozen model for reference. Models are composed of a generation body as well as a value head.
 superscript $^b$: batch, superscript $^{mb}$: mini-batch.
 $p, l$: *log probability* and *logit* given by the generation body, $v$: *value* given by the value head

1 **Procedure** Update(model, $x^b, y^b, R^b$):
   ▷ **Stage I:** forward passes to obtain reference stats on the **batch**
2  $\quad (p^b, l^b, v^b) \leftarrow$ BatchedForwardPass(model, $x^b, y^b$)
3  $\quad (p_r^b, l_r^b, v_r^b) \leftarrow$ BatchedForwardPass(ref_model, $x^b, y^b$)
4  $\quad s^b \leftarrow$ ComputeScores($R^b, p^b, p_r^b$)          ▷ compute the modified reward (Eq. 2)
   ▷ **Stage II:** update on **minibatches**
5  $\quad D^b \leftarrow (x^b, y^b, l^b, v^b, s^b)$
6  $\quad$ **for** $i = 1$ **to** $T_{PPO}$ **do**
7  $\qquad$ **for** $D^{mb} \in D^b$ **do**
8  $\qquad\quad (x^{mb}, y^{mb}, l^{mb}, v^{mb}, s^{mb}) \leftarrow D^{mb}$          ▷ take out a minibatch
9  $\qquad\quad (p, l, v) \leftarrow$ BatchedForwardPass(model, $x^{mb}, y^{mb}$)
10 $\qquad\quad$ TrainMinibatch(model, $p^{mb}, v^{mb}, s^{mb}, p, l, v$)          ▷ with PPO objective

   ▷ main loop
11 **for** $i = 1$ **to** $T$ **do**
   ▷ Take out a batch
12 $\quad$ **for** $x^b \in D$ **do**
13 $\qquad y^b \leftarrow$ model.generate($x^b$)                    ▷ obtain the model responses
14 $\qquad R^b \leftarrow r(x^b, y^b)$              ▷ obtain the rewards via the reward model $r$
15 $\qquad$ Update(model, $x^b, y^b, R^b$)
16 **return** model

---

## 5 PRIVATELY ALIGNING LMS WITHOUT HUMAN IN THE LOOP

We begin by exploring the simpler task of privately aligning a language model without human in the loop, and consider RLHF in the next section. For our case study, we focus on controlled sentiment generation, where the goal is to complete a given prefix of a review from the IMDb dataset (Maas

## Stage I: Supervised Fine-Tuning

Objective: learn to generate a review

## Stage II: RL Fine-Tuning

Objective: learn to generate a review with **positive** sentiment

Figure 1: **Case study: aligning a language model without human in the loop.** The goal is to complete a partial review with positive sentiment. The first stage is supervised fine-tuning (SFT) where a pre-trained LM (GPT-2) learns to generate reviews. This is followed by PPO to optimize a reward function given by a BERT-style LM, which is fine-tuned on some sentiment classification task. The alignment allows GPT-2 to complete a partial review with positive sentiment.

Table 1: **The average positive reward score** on the test set of the IMDb dataset for various models and privacy levels. $\epsilon = 0$ represents the pre-trained model. $\epsilon \in \{1, 2, 4, 8\}$ are privately aligned models with different privacy budgets. $\epsilon = \infty$ stands for alignment without any privacy. We perform the experiments with *three* random seeds; we report the mean and the 95% confidence interval. Additional privacy-utility trade-offs are demonstrated in Fig. 3 of Appendix C.2.

| Model | $\epsilon = 0$ | $\epsilon = 1$ | $\epsilon = 2$ | $\epsilon = 4$ | $\epsilon = 8$ | $\epsilon = \infty$ |
|---|---|---|---|---|---|---|
| GPT-2 | -0.30 | $1.47 \pm 0.81$ | $2.35 \pm 0.52$ | $2.74 \pm 0.27$ | $2.81 \pm 0.19$ | $3.10 \pm 0.22$ |
| GPT-2 Medium | -0.28 | $2.39 \pm 0.52$ | $2.60 \pm 0.43$ | $2.93 \pm 0.17$ | $2.93 \pm 0.13$ | $3.45 \pm 0.02$ |
| GPT-2 Large | -0.24 | $0.71 \pm 0.13$ | $1.91 \pm 0.42$ | $3.20 \pm 0.23$ | $3.38 \pm 0.03$ | $3.32 \pm 0.06$ |

et al., 2011)[1] with positive sentiment as depicted in Figure 1. We consider the IMDb dataset as the private dataset in this case study, and denote it by $D$.

As described in Section 2.1, this scenario consists of two steps. First, supervised fine-tuning (SFT) is performed on the pre-trained model $LM^{\text{pt}}$ with the language modeling objective, allowing it to achieve review generation capabilities. Then, we further fine-tune the model $LM^{\text{sft}}$ using PPO with the guidance from the reward model $r$, for the purpose of alignment towards generating reviews with a positive sentiment. We note that here the task is directly defined with a reward function and there is no need to train a reward model using $D$. Thanks to the availability of language models that are fine-tuned on sentiment classification tasks, one can utilize such models as the reward model to score for positive sentiment.

**Experimental setup.** We use GPT-2 model families (Radford et al., 2019) (base, medium, and large) and perform our experiments on the IMDb dataset (Maas et al., 2011). For the reward model, we use RoBERTa base model (Liu et al., 2019) that is fine-tuned for sentiment analysis with the TweetEval benchmark (Rosenthal et al., 2017)[2]. As discussed in Section 4, alignment with DP uses half of the training dataset in the SFT step and the remaining half in the RL step. Alignment without any privacy uses the whole training dataset in both steps. *Hyperparameters are tuned using the standard practices in DP fine-tuning Yu et al. (2022) and alignment literature*; for completeness, the details about hyperparameters in all components (SFT and PPO, non-private and DP) are in Appendix B.

**Evaluation.** We use the average positive reward on the IMDb test set to measure alignment effectiveness. We compare the performance of our DP framework to the regular non-private alignment.

**Main results.** We present the results in Table 1 for various privacy levels $\epsilon \in \{1, 2, 4, 8\}$ while fixing $\delta = 1/|D|$. We point out that these DP guarantees would also hold with smaller $\delta$, albeit with a minor increase of $\epsilon$ using privacy curves (Balle & Wang, 2018). Main highlights are:

1. We note that fully private ($\epsilon = 0$) pre-trained models are not aligned to generate positive reviews, as expected. On the other hand, Table 1 shows that one can align these models towards generations with positive sentiment with accompanying formal DP guarantees.

---

[1] https://huggingface.co/datasets/imdb
[2] https://huggingface.co/cardiffnlp/twitter-roberta-base-sentiment

Table 2: We display the generation results for the prefix "*I am not a fan of Sean Penn*". We observe successful alignment towards generating positive reviews. More results are in Appendix C.1.

| Model | $\epsilon = 4$ | $\epsilon = \infty$ |
|---|---|---|
| GPT-2 | *I am not a fan of Sean Penn* at all and I don't really look for him. I liked the flavour really | *I am not a fan of Sean Penn* and I love it. However, I became a bit too. I love the |
| GPT-2 Medium | *I am not a fan of Sean Penn*'s, I'm really happy and I love the movie, and I's very | *I am not a fan of Sean Penn*. I appreciate what he is. It's awesome. This has been amazing. |
| GPT-2 Large | *I am not a fan of Sean Penn* <3 this film is great and worth watching! <3 <3 <3 | *I am not a fan of Sean Penn*, but I love his work in baseball and I love his work for my favorite |

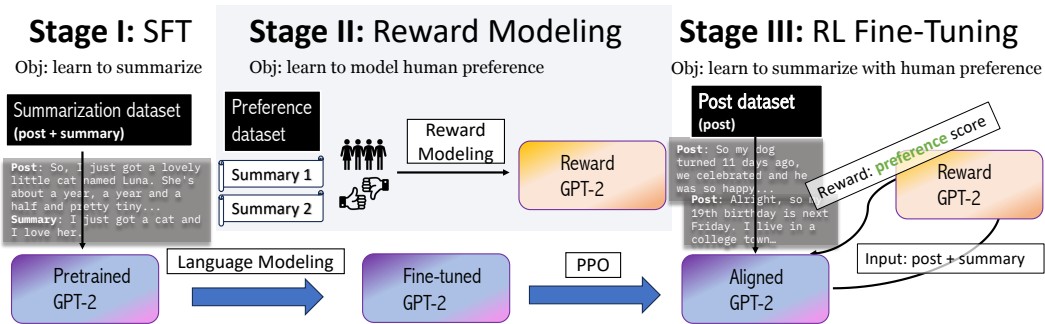

Figure 2: **Case study: aligning a language model with human preferences.** The goal is to generate the summary of a post in a human-preferred way. The first stage is supervised fine-tuning (SFT) where a pre-trained LM (GPT-2) learns to generate summaries. In the second stage the reward model is trained based on Eq. 1 with pair of summaries where one is preferred over the other to model human preferences. This is followed by PPO with the reward model from the second stage. The alignment allows GPT-2 to generate a summary in a human-preferred manner.

2. As expected, relaxing the privacy budget improves the average positive reward score on the test set and we observe strong performance at $\epsilon = 4$, which is commonly used in the DP fine-tuning literature (Yu et al., 2022).

3. Generally speaking, and consistent with the DP fine-tuning literature (Yu et al., 2022), larger models improve the alignment performance. One exception is GPT2-Large model for small $\epsilon$. The latter may be due to insufficient hyperparameter tuning as we tuned hyperparameters by fixing $\epsilon = 4$. For the non-private alignment ($\epsilon = \infty$), we observe a further improvement as expected as the privacy-utility trade-off is tilted completely in favor of utility.

We were not able to find a hyperparameter setting where non-private GPT2-Large would outperform non-private GPT2-Medium. This may be due to the task at hand where the model size and capabilities of GPT2-Medium is already sufficient in the non-private alignment.

**Demonstrations.** We randomly select five partial reviews from the test set and let the private ($\epsilon = 4$) and the non-private models complete the reviews. Part of the results are shown in Table 2 (more in Appendix C.1). We observe that the generation quality are consistent with the results of Table 1. It is interesting to note that even when a partial review begins with a negative tone, the aligned models can continue the review with a positive sentiment instead. Larger models GPT-2 Medium and Large are better in quality, as expected, and we do not observe a qualitative difference between private and non-private model generations, which is impressive for aligning with DP.

## 6 PRIVATELY ALIGNING LMS WITH HUMAN PREFERENCES

In this section we empirically evaluate the scenario where we privately align a language model with human preferences. For our case study, we focus on a summarization task, where the goal is to

Table 3: **The average reward score** (denoted by $r$) on the test set of the Reddit TL;DR dataset and **ROUGE-L score** (denoted by R-L) between model generated summaries and the label summaries for various models and privacy levels. $\epsilon = 0$ represents the pre-trained model. $\epsilon \in \{1, 2, 4, 8\}$ are privately aligned models with different privacy budgets. $\epsilon = \infty$ stands for alignment without any privacy. Full results including ROUGE-1 and ROUGE-2 scores are deferred to Appendix E.

| Model | $\epsilon = 0$ Pre-trained | | | $\epsilon = 1$ | | $\epsilon = 2$ | | $\epsilon = 4$ | | $\epsilon = 8$ | | $\epsilon = \infty$ | |
|---|---|---|---|---|---|---|---|---|---|---|---|---|---|
| | $r$ | R-L | | $r$ | R-L | $r$ | R-L | $r$ | R-L | $r$ | R-L | $r$ | R-L |
| GPT-2 | 0.05 | 8.26 | **SFT** | 0.44 | 11.45 | 0.48 | 11.84 | 0.50 | 12.30 | 0.49 | 12.45 | 0.63 | 14.48 |
| | | | **Aligned** | 0.22 | 10.41 | 0.53 | 11.44 | 0.68 | 12.33 | 0.69 | 11.74 | 1.53 | 14.17 |
| GPT-2 medium | 0.11 | 8.67 | **SFT** | 0.68 | 12.80 | 0.66 | 13.07 | 0.65 | 13.30 | 0.65 | 13.5 | 0.70 | 14.30 |
| | | | **Aligned** | 0.59 | 12.86 | 0.92 | 13.26 | 0.92 | 13.44 | 0.86 | 13.79 | 1.76 | 13.17 |
| GPT-2 large | -0.06 | 10.34 | **SFT** | 0.51 | 14.98 | 0.51 | 14.86 | 0.52 | 15.14 | 0.51 | 15.04 | 0.54 | 15.53 |
| | | | **Aligned** | 0.40 | 14.75 | 1.14 | 14.58 | 1.06 | 13.88 | 0.93 | 14.37 | 1.49 | 14.64 |

generate a summary of a post from the Reddit TL;DR summarization dataset (Völske et al., 2017) in a human-preferred manner as depicted in Figure 2. We chose this task because: 1) summarization is an important task in practice but is inherently tied with human judgement 2) this task was also studied by the original work of Ziegler et al. (2020) and their follow-up (Stiennon et al., 2022). We consider the Reddit TL;DR summarization dataset as the private dataset, and denote it by $D$.

Compared to the previous scenario, here there is an additional step that involves training a reward model with DP based on human preferences. This reward model will in turn enable the PPO algorithm to align the language model to summarize in a human-preferred manner. Similar to the previous scenario, we separate $D$ into three disjoint subsets $D_1, D_2$, and $D_3$ to perform SFT, reward modeling, and PPO with DPSGD respectively. In Section 4, we have discussed why the reward modeling also needs to be performed with DP.

**Experimental setup.** We use GPT-2 model families (Radford et al., 2019) (base, medium, and large). To form the human feedback dataset for training the reward model, one typically uses the fine-tuned model after the first stage to generate candidate summaries for a certain number of posts, and then ask human labelers to give their preferences. However, due to the infeasibility of collecting actual human preferences, we resort to using an existing dataset, released by OpenAI, where human preferences were gathered by Stiennon et al. (2022)[3]. The human feedback dataset in Stiennon et al. (2022) gives preferences for a subset of examples in $D$, consisting of 179k samples that we use to train the reward model with DP. Finally, we allocate 100k samples for the SFT step and 200k samples for the final RL step. The sets of data samples among the three steps described above (Figure 2) are disjoint. We provide the details about hyperparameters in Appendix D.

**Evaluation.** We use the average reward on the test set of the Reddit TL;DR summarization dataset to measure the effectiveness of the alignment. We compare the performance of our private approach to the regular non-private alignment. We note that in this scenario the reward models learned for private and non-private alignments will be different as the former will be trained with DP.

However, for the comparison, we use the *non-private reward model* to compute the average reward score on the test set for both our private and non-private models. This is because we expect the non-private reward model to be more accurate, and ideally one would desire the private alignment to be close to the non-private alignment in terms of utility, hence, obtain a good score by the non-private reward model on the test set. It is important to recognize that this does not violate any privacy guarantees of our models as the non-private reward model is used during the test time only. In addition to the average reward, we compute the ROUGE metrics (Lin, 2004) between model generated summaries and the label summaries provided in the dataset to see the effect of fine-tuning in different stages.

**Main results.** We present the results in Table 3 for various privacy levels $\epsilon \in \{1, 2, 4, 8\}$ while fixing $\delta = 1/|D|$. Main takeaways are:

---

[3]https://huggingface.co/datasets/openai/summarize_from_feedback/viewer/comparisons/train?row=0

1. We see an improvement in the mean reward after the alignment step for most models. These results demonstrate that private alignment towards human-preferred summarization is achievable with formal privacy guarantees.
2. Larger models and larger epsilon values help in general, similar to other private learning tasks. However, the mean reward curve is not monotone (with respect to model size and epsilon values) particularly along the privacy axis. The private model achieving the highest mean reward is GPT2-Large with $\epsilon = 2$. More extensive hyperparmeter tuning is necessary to understand this phenomenon.
3. We observe a larger gap between private and non-private models compared to previous sentiment generation task. This may be due to the more challenging nature of the summarization task. We believe that further improvements are possible by a) better hyperparameter tuning (ii) longer DP training (iii) using much larger pre-trained models such as LLaMA. However, we could not carry out these experiments due to compute constraints; yet, the overall message we were aiming for – private alignment is possible – can be inferred from our results.
4. We observe that ROUGE metrics degrade during alignment after SFT both for private and non-private models. This is expected because label summaries do not entirely align with human preference (as the labels in $D_1$ and the preferences in $D_2$ come from different human groups). Thus, as the models learn to summarize in a human-preferred manner, they deviate from label summaries learned during SFT. Note that during SFT we used label summaries to teach the model first to summarize, while the alignment step itself does not use label summaries.

## 7 RELATED WORK

**Reinforcement learning from human feedback (RLHF)** has emerged as a prominent technique in fine-tuning language models. Christiano et al. (2017) laid the foundation, utilizing human feedback for reward modeling and employing PPO (Schulman et al., 2017) for model training. Early applications of RLHF in the natural language realm focused on stylistic continuation (Ziegler et al., 2020), summarization (Ziegler et al., 2020; Stiennon et al., 2022; Wu et al., 2021), etc. Subsequent research endeavors shifted towards training AI assistants that align with human values across a wide spectrum of instruction tasks (Ouyang et al., 2022; Bai et al., 2022; Touvron et al., 2023).

**DP in language models** Exploiting the memorization ability of language models (Carlini et al., 2023), privacy attacks have been launched, extracting training data or inferring membership (Carlini et al., 2019; 2021; Elmahdy et al., 2022; Mattern et al., 2023). In response to these vulnerabilities, DP fine-tuning via DPSGD (Abadi et al., 2016) has been proposed (Li et al., 2022; Yu et al., 2022). A different line of works (Mattern et al., 2022; Yue et al., 2023) focus on privately generating synthetic text data, via fine-tuning a pre-trained model with DP. Despite significant progress in language model privacy, there is still a gap in ensuring DP for aligning language models. To our best knowledge, we are the first that take a step in this direction.

**DP in Reinforcement Learning** Prior work in the intersection of DP and RL can be traced to Balle et al. (2016). Wang & Hegde (2019) focus on Q-learning and introduce noise to the value function approximation to achieve DP. Ma et al. (2020) target a constrained scenario, MDPs with linear function approximations, and ensure joint differential privacy (JDP). Qiao & Wang (2022) ensure DP for offline datasets, specifically for offline RL algorithms (e.g., APVI (Yin & Wang, 2021)). None of these fulfills the need of achieving DP for online RL (e.g., PPO) with the neighboring relation defined on a fixed dataset. Our DP adaptation of PPO (Section 4) fills the gap.

We defer a more complete description of the related work to Appendix H.

## 8 CONCLUSIONS AND FUTURE WORK

In this paper we initiated the study of privately aligning LLMs with human feedback. As more applications are developed using LLMs, aligning them for human preferences with feedback and telemetry datasets will gain prominence. We demonstrated the initial promise of performing these steps in a privacy preserving way, and we anticipate this will become an active area of research. Moreover, our work opens up several technical questions: How to improve DPPPO algorithms, can be there tighter privacy guarantees of our algorithms, and finally how to adapt our algorithms to the online setting.

ACKNOWLEDGEMENT

Fan Wu would like to thank Yuzheng Hu for his support. Bubble—Fan's beloved stuffed bunny—would like to thank Microsoft Research for the lovely campus and environment.

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

## A  DPSGD ALGORTIHM

We provide a pseudocode for the DPSGD algorithm in Algorithm 2. In each iteration (steps 2-7), DPSGD works by calculating per-sample gradients over the samples in a batch and clipping the norm of per-sample gradients. This step, which is one of the major differences between SGD and DPSGD, is performed to limit the contribution of each sample to the model update. Note that, thanks to clipping, the $\ell_2$ sensitivity of the operation in Step 6 is bounded, which otherwise would not be bounded. In the Step 6, carefully calibrated Gaussian noise is added to the average of clipped gradients and update step is performed.

The privacy analysis of DPSGD works as follows. Fix one iteration of the algorithm. Since the clipping step ensures that the $\ell_2$-sensitivity of the average of gradients remains bounded, it is not hard to prove that each iteration of DPSGD satisfies $(\epsilon, \delta)$-DP with some privacy parameters. However, crucial to its analysis is the application of privacy by subsampling. Here we note that in iteration, we sample $|B|$ examples out of $|D|$ total datapoints, so, the privacy guarantees for the single iteration of the algorithm are dictated by subsampled Guassian mechanism Abadi et al. (2016); Gopi et al. (2021). Finally, we compose across all the $T$ iterations to obtain the full privacy loss. The PRV account that we use Gopi et al. (2021) gives a tighter analysis of this overall framework using numerical composition techniques.

---

**Algorithm 2:** Differential Privacy Stochastic Gradient Descent (DPSGD)

---

**Define:** Dataset $D$, model parameters $\theta$, loss function $\mathcal{L}(\theta, x)$, learning rate $\eta$, noise scale $\sigma$,
gradient norm bound $C$, sampling probability $p$, number of epochs $T$

1  **for** $t = 1, 2, \ldots, T$ **do**
2      Sample $B \subseteq D$ with sampling probability $p$
3      **for** $x_i \in B$ **do**
4          Compute gradient: $g_i \leftarrow \nabla_\theta \mathcal{L}(\theta, x_i)$
5          Clip gradient: $g_i \leftarrow g_i / \max(1, \frac{\|g_i\|_2}{C})$
6      Add noise and calculate update: $g \leftarrow \frac{1}{|B|} \left( \sum_i g_i + \mathcal{N}(0, \sigma^2 C^2 \mathbf{I}) \right)$
7      Update model: $\theta \leftarrow \theta - \eta \cdot g$
8  **return** $\theta$

---

## B  HYPERPARAMETERS FOR SECTION 5

In the following, we describe the details of our hyperparameter search for the results in Section 5.

For LoRA, we choose the bottleneck rank $r = 4$ and fine-tune query and value matrices of the attention layers as in the original paper (Hu et al., 2022).

For non-private SFT, we tune the batch size and the learning rate from the set $\{8, 16, 32, 64\}$ and in the range [1e-6, 1e-2] respectively. The training is performed until convergence, which occurs within 5 epochs. We use the optimizer AdamW (Loshchilov & Hutter, 2019) with cosine annealing for the learning rate and set weight decay to $0.01$. The final batch size and learning rate are reported in Table 4.

Table 4: Non-private SFT hyperparameters for the results in Section 5.

| Model | Batch size | Learning rate |
|---|---|---|
| GPT-2 | 64 | 5e-4 |
| GPT-2 Medium | 64 | 5e-4 |
| GPT-2 Large | 64 | 2e-4 |

For DP SFT, informed by prior work (Yu et al., 2022; Li et al., 2022), we aim to set large batch size and constant learning rate with a long training course. We set the batch size to 512 and the number of epochs to 40. We similarly tune the learning rate in the range [1e-5, 1e-1] and finally set to 3e-4

for all models. We use the optimizer AdamW with weight decay $0.01$. For the DP parameters, we set a small per-sample clipping norm as $1.0$ and calculate the corresponding noise multiplier to achieve the reported $(\epsilon, \delta)$-DP using the accountant in Gopi et al. (2021).

For PPO, we use the TRL framework[4] and set the hyperparameters specific to PPO as default values therein. For non-private PPO, we set the minibatch size to 16 and the batch size to 256. PPO epochs is set to 4 and one epoch is passed on the full dataset. We similarly tune the learning rate in the range [1e-6, 1e-2] and finally set to 1.4e-3 for GPT-2 and GPT-2 Medium, and 2e-4 for GPT-2 Large.

For DPPPO, we follow a similar course as DP SFT. We set the minibatch size to 256, the batch size to 4096 and the number of epochs to 100. PPO epochs must be set to 1 as explained in Section 5. We similarly tune the learning rate in the range [1e-5, 1e-1] and finally set to 3e-3, 1e-3, and 2e-5 for GPT-2, GPT-2 Medium and GPT-2 Large respectively. DP parameters also follow as DP SFT.

### B.1 ABLATION STUDY ON $T_{\mathrm{PPO}}$

We perform an ablation study on $T_{\mathrm{PPO}}$ using the GPT-2 model for $\epsilon = 4$ to investigate the implications of setting $T_{\mathrm{PPO}} = 1$ in our DPPPO algorithm. We report the results in Table 5. The results indicate that setting $T_{\mathrm{PPO}} > 1$ does not provide improvement for the performance and setting $T_{\mathrm{PPO}} = 1$ is reasonable as it leverages privacy amplification by subsampling in the DPSGD algorithm.

Table 5: **Ablation study on $T_{\mathrm{PPO}}$.** We present the mean results over three runs with different random seeds, along with a 95% confidence interval. Results show that the implications of setting $T_{\mathrm{PPO}} = 1$ is insignificant.

| Model | $\epsilon$ | $T_{\mathrm{PPO}}$ | Average reward |
|---|---|---|---|
| GPT-2 | 4 | 1 | $2.74 \pm 0.27$ |
| | | 2 | $2.72 \pm 0.14$ |
| | | 4 | $2.73 \pm 0.05$ |
| | | 8 | $2.64 \pm 0.81$ |

## C  ADDITIONAL RESULTS FOR THE POSITIVE REVIEW GENERATION TASK IN SECTION 5

We present the following additional results as a compliment to Table 1 in Section 5.

### C.1 SAMPLE GENERATIONS FOR SECTION 5

Table 6 demonstrates the alignment towards generation with positive sentiment for private and non-private models via completions on randomly sampled prefixes from the test set.

### C.2 TRADE-OFF BETWEEN PRIVACY AND UTILITY

To provide a clearer understanding of the privacy-utility trade-off, we illustrate in Figure 3 how different levels of privacy (varying $\epsilon$) impact the model's performance for the GPT-2 Medium model. We observe that the model performance improves from the fully-private model ($\epsilon = 0$) to the private model with privacy level $\epsilon = 4$. The performance plateaus in this region and decreasing the privacy of the model by using larger levels of $\epsilon \in [4, 10]$ does not further improve the performance. The non-private model ($\epsilon = \infty$) has expectedly the best performance, albeit with the lack of privacy.

## D  HYPERPARAMETERS FOR SECTION 6

We mostly follow the hyperparameters described in Appendix B. Here we state only the differences.

---

[4]https://huggingface.co/docs/trl/index

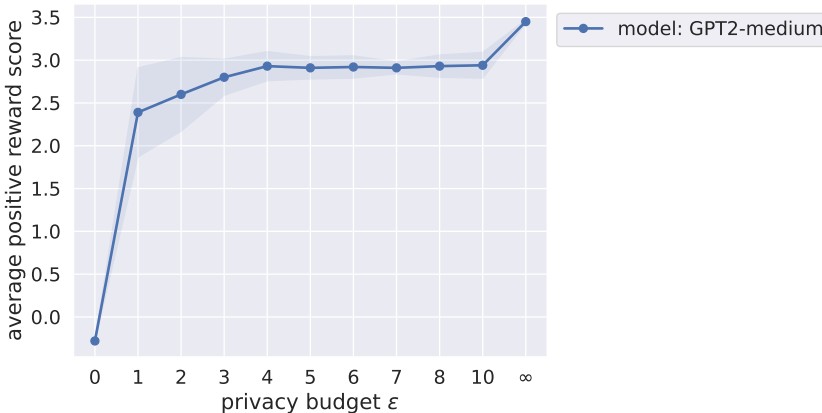

Figure 3: **Trade-off between utility and privacy for the positive review generation task.** Results are obtained on the GPT2-medium model. The shaded area denotes the 95% confidence interval. $\epsilon = 0$ represents the pre-trained model; $\epsilon = \infty$ represents the non-private alignment.

Compared to the scenario in Section 5 we work with an order of magnitude larger dataset size in this scenario. Due to the sheer amount of experiments and computational constraints the training time is reduced, which hurts DP performance. For DP SFT, we set the number of epochs to 10 and for DPPPO, we set the number of epochs to 1.

An important difference is that this scenario involves training a reward model. We fix GPT-2 model to be used for reward model in all experiments. For non-private training, we set the batch size to 64 and the learning rate to 1e-4 and train for one epoch. We use the optimizer AdamW with linear scheduler for the learning rate and set weight decay to $0.01$. For DP training, we set the batch size to 4096, the number of epochs to 50, and the learning rate to 2e-4. We use the optimizer AdamW with weight decay $0.01$. For the DP parameters, we set a small per-sample clipping norm as $1.0$ and calculate the corresponding noise multiplier to achieve the reported $(\epsilon, \delta)$-DP using the accountant in Gopi et al. (2021).

## E  FULL RESULTS FOR THE SUMMARIZATION TASK IN SECTION 6

We present the complete set of results for the summarization task in Table 7, additionally including the ROUGE-1 and ROUGE-2 scores.

## F  FULL PSEUDO-CODE

We present the complete version of the pseudo-code in Algorithm 3. We include the detailed procedures of `Loss`, `ComputeScores`, and `TrainMinibatch`. The parts that require additional adaptation to fulfill DP are highlighted in blue and red.

## G  TWO PARADIGMS OF ALIGNING LANGUAGE MODELS

Depending on the nature of the reward signal—whether it is from some standard and commonly endorsed criteria or from the preferences from a group of humans, there are two main paradigms in using RL for alignment.

**RL without human in the loop.**   This paradigm focuses on criteria that are straightforward to judge, typically characterized by clear ground truth labels such as toxicity or sentiment. Given their easily quantifiable nature, these criteria often align with binary labels. Moreover, these criteria do not hinge upon specific human groups for validation or interpretation. The advantage of this

---

**Algorithm 3:** Aligning language models with RL (PPO), full version

---

**Define:** $D$: a dataset consisting of input texts. $x$: input text, $y$: model response.

$T$: total training epochs, $T_{\text{PPO}}$: PPO training epochs.

model, ref_model: the model being learned and the frozen model for reference.

Models are composed of a generation body as well as a value head.

superscript $^b$: batch, superscript $^{mb}$: mini-batch.

$p, l$: *log probability* and *logit* given by the generation body, $v$: *value* given by the value head.

1 **Function** Loss$(p^{old}, v^{old}, s^{old}, p, l, v)$**:**

2 $\quad A \leftarrow$ ComputeAdvantages$(v^{\text{old}}, s^{\text{old}})$ ▷ `through generalized advantage` `estimation (Schulman et al., 2015)`

3 $\quad r \leftarrow \exp(p - p^{\text{old}})$ ▷ `compute the ratio`

4 $\quad \text{loss}_p \leftarrow \min(-rA, -\text{Clip}(r, 1 - \varepsilon, 1 + \varepsilon)A)$ ▷ `clipped objective`

5 $\quad \text{loss}_v \leftarrow \alpha_v \cdot (A + v^{\text{old}} - v)^2.$`mean()`

6 $\quad$ **return** $\text{loss}_p, \text{loss}_v$

7 **Function** ComputeScores$(R^b, p^b, p_r^b)$**:**

$\quad$ ▷ `adjust the score by KL divergence. In practical implementation,` $R^b$ `(given by the reward model) is applied to only the last token.`

8 $\quad$ **return** $R^b - \alpha_{\text{KL}} \cdot (p^b - p_r^b)$

9 **Procedure** TrainMinibatch$(\text{model}, p^{old}, v^{old}, s^{old}, p, l, v)$**:**

10 $\quad \text{loss}_p, \text{loss}_v \leftarrow$ Loss$(p^{\text{old}}, v^{\text{old}}, s^{\text{old}}, p, l, v)$

11 $\quad \text{loss} = \text{loss}_p + \text{loss}_v$ ▷ `sum of policy loss and value loss`

12 $\quad$ optimizer.`zero_grad()`

13 $\quad$ loss.`backward()`

14 $\quad$ optimizer.`step()`

15 **Procedure** Update$(\text{model}, x^b, y^b, R^b)$**:**

$\quad$ ▷ **Stage I:** forward passes to obtain reference stats on the **batch**

16 $\quad (p^b, l^b, v^b) \leftarrow$ BatchedForwardPass$(\text{model}, x^b, y^b)$

17 $\quad (p_r^b, l_r^b, v_r^b) \leftarrow$ BatchedForwardPass$(\text{ref\_model}, x^b, y^b)$

18 $\quad s^b \leftarrow$ ComputeScores$(R^b, p^b, p_r^b)$ ▷ `compute the modified reward (Eq. 2)`

$\quad$ ▷ **Stage II:** update on **minibatches**

19 $\quad D^b \leftarrow (x^b, y^b, l^b, v^b, s^b)$ ▷ `compose batched data`

20 $\quad$ **for** $i = 1$ **to** $T_{PPO}$ **do**

21 $\quad\quad$ **for** $D^{mb} \in D^b$ **do**

22 $\quad\quad\quad (x^{mb}, y^{mb}, l^{mb}, v^{mb}, s^{mb}) \leftarrow D^{mb}$ ▷ `take out a minibatch`

23 $\quad\quad\quad (p, l, v) \leftarrow$ BatchedForwardPass$(\text{model}, x^{mb}, y^{mb})$

24 $\quad\quad\quad$ TrainMinibatch$(\text{model}, p^{mb}, v^{mb}, s^{mb}, p, l, v)$ ▷ `with PPO objective`

25

$\quad$ ▷ `main loop`

26 **for** $i = 1$ **to** $T$ **do**

$\quad$ ▷ `take out a batch`

27 $\quad$ **for** $x^b \in D$ **do**

28 $\quad\quad y^b \leftarrow \text{model.generate}(x^b)$ ▷ `obtain the model responses`

29 $\quad\quad R^b \leftarrow r(x^b, y^b)$ ▷ `obtain the rewards via the reward model r`

30 $\quad\quad$ Update$(\text{model}, x^b, y^b, R^b)$

31 **return** model

---

paradigm is that there exists a plethora of pre-trained classifiers[5] and detection APIs[6] available to

---

[5]`https://huggingface.co/nlptown/bert-base-multilingual-uncased-sentiment`, `https://huggingface.co/cardiffnlp/twitter-roberta-base-sentiment-latest`

[6]`https://developers.perspectiveapi.com/s/about-the-api?language=en_US`

the public. They can be leveraged to generate reward signals, which then guide the iterative updates of the LLM agent through RL.

**RL with human preferences.** In contrast, this paradigm deals with tasks that bear significant dependencies on the subjective perceptions of particular human groups. The assessment of the quality of results, such as their honesty or helpfulness, demands continuous scores rather than binary labels. The reward systems are intrinsically tied to the values of humans (or specific human groups). Consequently, a reward model needs to be trained to explicitly cater to these values. After training the reward model to capture human preferences, it is incorporated into the RL process to guide the LLM agent in adopting these preferences.

## H  FULL VERSION OF THE RELATED WORK

**Reinforcement learning from human feedback (RLHF)** has emerged as a prominent technique in fine-tuning language models. Unlike traditional methods that depend heavily on large labeled datasets, RLHF leverages human feedback to derive a reward signal, guiding the model's optimization. This enables models to produce more desired outputs in complex and open-ended tasks. Christiano et al. (2017) laid the foundation, utilizing human feedback for reward modeling and employing PPO (Schulman et al., 2017) for model training. Early applications of RLHF in the natural language realm focused on stylistic continuation (Ziegler et al., 2020), summarization (Ziegler et al., 2020; Stiennon et al., 2022; Wu et al., 2021), and translation (Nguyen et al., 2017; Kreutzer et al., 2018). Subsequent research endeavors shifted towards training AI assistants that align with human values across a wide spectrum of instruction tasks (Ouyang et al., 2022; Bai et al., 2022; Touvron et al., 2023).

**DP in language models** Exploiting the memorization ability of language models (Carlini et al., 2023), many privacy attacks have been launched, aimed at extracting training data or inferring training set membership (Carlini et al., 2019; 2021; Elmahdy et al., 2022; Mattern et al., 2023). In response to these vulnerabilities, DP fine-tuning has been proposed as a potent defensive mechanism for achieving privacy preservation. Li et al. (2022); Yu et al. (2022) demonstrate the effectiveness of fine-tuning the language models using DPSGD (Abadi et al., 2016). Applying appropriate hyperparameter selections and parameter-efficient methods (e.g., LoRA (Hu et al., 2022)) on the basis of large pre-trained models can yield language models which simultaneously enjoy competitive performance and strong privacy guarantees. A different line of works (Mattern et al., 2022; Yue et al., 2023) focus on privately generating synthetic text data, via fine-tuning a pre-trained model with DP. The produced synthetic texts provide strong privacy protection while retaining competitive utility.

Despite these substantial progresses in ensuring privacy for language model related applications, there remains a gap in ensuring DP for aligning language models. To our best knowledge, we are the first that take a step in this direction.

**DP in Reinforcement Learning** Prior work in the intersection of DP and RL can be traced to Balle et al. (2016). Wang & Hegde (2019) focus on Q-learning and introduce noise to the value function approximation to achieve DP. Ma et al. (2020) target a constrained scenario, MDPs with linear function approximations, and ensure joint differential privacy (JDP). Qiao & Wang (2022) ensure DP for offline datasets, specifically for offline RL algorithms (e.g., APVI (Yin & Wang, 2021)). None of these fulfills the need of achieving DP for online RL (e.g., PPO) with the neighboring relation defined on a fixed dataset. Our DP adaptation of PPO (Section 4) fills the gap.

Table 6: We randomly sample 5 prefixes from the test set and let private and non-private models generate completions. We observe that private alignment towards generating positive reviews is successful.

| Prefix | Model | $\epsilon = 4$ | $\epsilon = \infty$ |
|---|---|---|---|
| I loathe, despise, | GPT-2 | I loathe, despise, love eep too great ideas and functions perfect | I loathe, despise, and part of joined in and is still handled |
| | GPT-2-M | I loathe, despise, love and I love this game, it's | I loathe, despise, but I love this book. Hats! And |
| | GPT-2-L | I loathe, despise, love this movie! I was really happy! | I loathe, despise, love us. I love us! I want |
| Seriously! You've just got to see | GPT-2 | Seriously! You've just got to see this awesome comedy! It is fun funny | Seriously! You've just got to see this so what wonderful stuff we're going |
| | GPT-2-M | Seriously! You've just got to see it! I am very appreciative of | Seriously! You've just got to see watching this cool movie. The movie is |
| | GPT-2-L | Seriously! You've just got to see this awesome movie!! It's awesome! | Seriously! You've just got to see this beautiful collection. We love the way |
| With a title like that, you | GPT-2 | With a title like that, you will love it! I love this. It is exciting and could make it really | With a title like that, you have huge up and great. It is a fantastic story and I enjoyed it all |
| | GPT-2-M | With a title like that, you can't help but feel positive but certainly is a very inspiring concept and the way | With a title like that, you're amazing, we're ready to continue. It looks cooler. I can't |
| | GPT-2-L | With a title like that, you know special production...great job!! Jessica is great! Great material and great acting | With a title like that, you're right. I love this site! It makes me feel good, and I |
| I am not a fan of Sean Penn | GPT-2 | I am not a fan of Sean Penn at all and I don't really look for him. I liked the flavour really | I am not a fan of Sean Penn and I love it. However, I became a bit too. I love the |
| | GPT-2-M | I am not a fan of Sean Penn's, I'm really happy and I love the movie, and I's very | I am not a fan of Sean Penn. I appreciate what he is. It's awesome. This has been amazing. |
| | GPT-2-L | I am not a fan of Sean Penn <3 this film is great and worth watching! <3 <3 <3 | I am not a fan of Sean Penn, but I love his work in baseball and I love his work for my favorite |
| In the original French version, the jokes | GPT-2 | In the original French version, the jokes were pretty fun and pretty neat. I really liked | In the original French version, the jokes are amazing. I love them so much, I |
| | GPT-2-M | In the original French version, the jokes are beautifully clear and funny. I am a very | In the original French version, the jokes are great, but I am excited to look at |
| | GPT-2-L | In the original French version, the jokes were very funny! my main pleasure from this movie | In the original French version, the jokes were quite good and it was quite close to the |

Table 7: The average reward score (denoted by $r$) on the test set of the Reddit TL;DR summarization dataset and ROUGE metrics (ROUGE-1, ROUGE-2, and ROUGE-L denoted by R-1, R-2, and R-L, respectively) between model generated summaries and the label summaries in the test set for various models and privacy levels. $\epsilon = 0$ represents the pre-trained model. $\epsilon \in \{1, 2, 4, 8\}$ are privately aligned models with different privacy budgets. $\epsilon = \infty$ is the alignment procedure without any privacy. Our results demonstrate that alignment towards human-preferred summarization is obtainable with formal privacy guarantees to the underlying dataset. Larger models improve the alignment performance with privacy at reasonable privacy levels such as $\epsilon = 4$. ROUGE metrics indicate that models can deviate from label summaries learned during SFT and align towards human-preferred summaries with PPO during alignment.

| Model | $\epsilon$ | Stage | Mean Reward | R-1 | R-2 | R-L |
|---|---|---|---|---|---|---|
| | 0 | Pre-trained | 0.05 | 12.91 | 0.78 | 8.26 |
| | 1 | SFT | 0.44 | 16.69 | 1.69 | 11.45 |
| | | Aligned | 0.22 | 14.69 | 1.50 | 10.41 |
| | 2 | SFT | 0.48 | 17.23 | 1.85 | 11.84 |
| GPT-2 | | Aligned | 0.53 | 16.62 | 1.53 | 11.44 |
| | 4 | SFT | 0.50 | 17.84 | 2.02 | 12.30 |
| | | Aligned | 0.68 | 17.75 | 1.80 | 12.33 |
| | 8 | SFT | 0.49 | 17.89 | 2.01 | 12.45 |
| | | Aligned | 0.69 | 16.55 | 1.62 | 11.74 |
| | $\infty$ | SFT | 0.63 | 20.85 | 2.97 | 14.48 |
| | | Aligned | 1.53 | 20.61 | 3.13 | 14.17 |
| | 0 | Pre-trained | 0.11 | 13.53 | 0.90 | 8.67 |
| | 1 | SFT | 0.68 | 18.70 | 2.36 | 12.80 |
| | | Aligned | 0.59 | 18.44 | 2.44 | 12.86 |
| | 2 | SFT | 0.66 | 18.79 | 2.47 | 13.07 |
| GPT-2 | | Aligned | 0.92 | 19.60 | 2.34 | 13.26 |
| Medium | 4 | SFT | 0.65 | 19.27 | 2.62 | 13.30 |
| | | Aligned | 0.92 | 19.48 | 2.45 | 13.44 |
| | 8 | SFT | 0.65 | 19.62 | 2.62 | 13.50 |
| | | Aligned | 0.86 | 19.85 | 2.65 | 13.79 |
| | $\infty$ | SFT | 0.70 | 20.59 | 2.85 | 14.30 |
| | | Aligned | 1.76 | 19.64 | 2.50 | 13.17 |
| | 0 | Pre-trained | -0.06 | 16.13 | 1.56 | 10.34 |
| | 1 | SFT | 0.51 | 21.67 | 3.37 | 14.98 |
| | | Aligned | 0.40 | 21.17 | 3.28 | 14.75 |
| | 2 | SFT | 0.51 | 21.41 | 3.35 | 14.86 |
| GPT-2 | | Aligned | 1.14 | 21.33 | 3.33 | 14.58 |
| Large | 4 | SFT | 0.52 | 21.83 | 3.47 | 15.14 |
| | | Aligned | 1.06 | 19.63 | 2.83 | 13.88 |
| | 8 | SFT | 0.51 | 21.71 | 3.34 | 15.04 |
| | | Aligned | 0.93 | 20.26 | 3.04 | 14.37 |
| | $\infty$ | SFT | 0.54 | 22.22 | 3.58 | 15.53 |
| | | Aligned | 1.49 | 21.81 | 3.32 | 14.64 |

