# OpenReview forum: "Privately Aligning Language Models with Reinforcement Learning"
_ICLR.cc/2024/Conference — ICLR 2024 poster_

### Official Review · Reviewer_JeaX · 2023-10-25

**Soundness:** 3 good
**Presentation:** 3 good
**Contribution:** 2 fair
**Rating:** 6
**Confidence:** 2

**Summary:**

The paper focuses on aligning large language models (LLMs) through reinforcement learning (RL) while preserving user privacy using Differential Privacy (DP). The paper introduces a new DP framework for this alignment and validates its effectiveness through experiments.

**Strengths:**

1. The paper proposes a differentially private framework for aligning LLMs with RL, offering mathematical guarantees of privacy.
2. The paper empirically evaluates the framework on tasks like positive review generation and summarization, showing that it offers competitive utility while ensuring strong privacy protections.

**Weaknesses:**

1. The paper employs DPSGD to ensure privacy in the alignment of large language models through reinforcement learning. While the use of DPSGD is well-established in the privacy literature. Furthermore, the paper does not introduce significant modifications to the RLHF process. The innovation seems to be more focused on engineering adjustments rather than novel theoretical contributions.

2. The paper discusses the trade-offs between privacy and utility but does not present these results in an intuitive manner. A Pareto frontier could be more illustrative in showing how different levels of privacy (varying ε) impact the model's performance. This would provide a clearer understanding of the trade-offs involved.

3. If the reward in step 2 is DP, is it necessary to use the DPPPO in step 3 as the learning reward is already DP?

**Questions:**

Please check the weaknesses.

---

> ### Author Response · Authors · 2023-11-21
> **Official response to reviewer JeaX**
>
> We thank the reviewer for their careful reading and thoughtful comments. We address all their questions below.
>
> > Q: The paper employs DPSGD to ensure privacy in the alignment of large language models through reinforcement learning. While the use of DPSGD is well-established in the privacy literature. Furthermore, the paper does not introduce significant modifications to the RLHF process. The innovation seems to be more focused on engineering adjustments rather than novel theoretical contributions.
>
> A: We acknowledge your observation regarding the use of DPSGD in our work and the perceived emphasis on engineering adjustments rather than novel theoretical contributions. While it's true that DPSGD is employed as a key component, we believe the primary contribution of our paper lies in the comprehensive framework we present for integrating differential privacy into the RLHF process for large language models (LLMs). Our approach highlights the necessity of ensuring each step is differentially private, discusses the adjustments needed for PPO, and examines their performance implications. These insights, we argue, are valuable algorithmic contributions to the field.
>
> Moreover, our paper underscores two significant conceptual messages for the broader DP and LLM community. 1) RLHF is an important step in training LLMs, but it can also lead to privacy concerns if not done with care. So, these are important problems to study and our work opens up new research questions for the DP community to explore. 2) By establishing a benchmark for DP RLHF, our work demonstrates its feasibility and invites further research and development in this area. We believe that such contributions are as important as contributions that show algorithmic improvements, and the line between engineering and research is quite thin when it comes to LLMs.
>
> > Q: The paper discusses the trade-offs between privacy and utility but does not present these results in an intuitive manner. A Pareto frontier could be more illustrative in showing how different levels of privacy (varying ε) impact the model's performance. This would provide a clearer understanding of the trade-offs involved.
>
> A: We thank the reviewer for the suggestion regarding the presentation of the trade-offs between privacy and utility. Following your recommendation, we have illustrated in Figure 3 of Appendix C in the revised manuscript how different levels of privacy (varying $\epsilon$) impact the model's performance for the GPT-2 Medium model. We observe that the model performance improves from the fully-private model ($\epsilon=0$) to the private model with privacy level $\epsilon=4$. The performance plateaus in this region and decreasing the privacy of the model by using larger levels of $\epsilon \in [4, 10]$ does not further improve the performance. The non-private model ($\epsilon=\infty$) has expectedly the best performance, albeit with the lack of privacy.
>
> > Q: If the reward in step 2 is DP, is it necessary to use the DPPPO in step 3 as the learning reward is already DP?
>
> A: Answer to this question depends on the threat model, and so let us recall the third step of the alignment process. In this step, a new prompt (query) is sampled from the user dataset, and the model generates multiple outputs to the query. The reward model then assigns a reward to each generated output, and we use PPO to update the model/policy. Note that this policy gradient step involves changing the weights of the LLM. Thus, if we assume that the query/prompt is private information, then, updating the weights of the model needs to be private as well. If the user queries are not considered private, then perhaps, we do not need this step to be DP. However, we anticipate that in numerous applications, user prompts are likely to be considered private information.
>
> We hope that this answers your question.
>
> We would be happy to engage in further discussions with the reviewer should any follow-up suggestions arise. We would also very much appreciate the reviewer considering increasing their rating in case they find our responses compelling.

---

> > ### Comment · Reviewer_JeaX · 2023-11-22
> >
> > Thank you for your reply. I appreciate your efforts in addressing my concerns during the review process.

---

### Official Review · Reviewer_vWsq · 2023-10-25

**Soundness:** 3 good
**Presentation:** 3 good
**Contribution:** 3 good
**Rating:** 6
**Confidence:** 3

**Summary:**

The authors provide privacy-preserving technique for fine-tuning large language models. They apply differentiall-private SGD (DP-SGD) to the PPO reinforcement learning algorithm during model fine tuning. They demonstrate that for single-digit privacy budgets it is possible to fine tune GPT-2 such that there is improved utility on positive reward score.

**Strengths:**

The paper combines two well-understood algorithms, DP-SGD and PPO in a well-motivated task of reinforcement with human feedback and supervised fine-tuning. The experiments and the results are clear and the application of private model fine-tuning is reasonable.

The paper is well organized and the writing is clear. The paper is well written and the algorithm is clearly explained. Their claims are based on the GPT-2 family of models and run experiments on the ROUGE metrics and the TweetEval benchmark. The authors offer a privacy-preserving technique to undertake reinforcement learning with human feedback. They combine DP-SGD and PPO with a few adaptations and show utility benefits on NLP benchmarks.

**Weaknesses:**

While the examples are helpful, the overall motivation could be a bit stronger. What are we protecting and why? What is the threat model around incorporating human feedback? Are their examples of memorization from human feedback?

The experiments in the main body do not include error or number of trials details. It is unclear in Table 1 why models with less provacy should do worse than those with more privacy (GPT-2 Medium, eps 4->8, or GPT-2 Large eps 8->Inf). Such results demand further study and/or ablations and are difficult to interpret without confidence intervals. The use of corporate imagery (Reddit / OpenAI) weakens the overall presentation and the generality of the results. Work in differential privacy and RL can be traced to differentially-private policy evaluation (Balle, Gomrockchi, Precup). The paper touches on the privacy accounting implications when  $T_{\text{PPO}} \neq 1$
, but does not offer evaluate the implication of fixing it to the default value of 4.

**Questions:**

What are the confidence intervals for your reported experiments?
Are there other domains where the fine-tuning method can be better understood?
What are we losing by setting $T_{\text{PPO}} \neq 1$ instead of 4 in terms of utility? Is this trade-off significant?

---

> ### Author Response · Authors · 2023-11-21
> **Official response to reviewer vWsq (1/2)**
>
> We thank the reviewer for their careful reading and thoughtful comments. We address all their questions below.
>
>
> > Q: While the examples are helpful, the overall motivation could be a bit stronger. What are we protecting and why? What is the threat model around incorporating human feedback? Are their examples of memorization from human feedback?
>
>
> A: Thanks for your question, and we agree that motivation could have been stronger. We will improve this in the future versions.
>
>
> Let us focus on your question regarding the threat model and examples of memorization from human feedback first note that in the alignment. To answer this, we note that the first step of the alignment process simply fine-tunes on the user demonstrations. We already know from previous works (Duan et al., 2023; Carlini et al.,2021; 2019) that this step can lead to LLMs memorizing significant user content. From a practical deployment perspective, this could be the most privacy-sensitive step of the RLHF process, although rest of the steps needs to be DP to satisfy that overall framework is provably DP.
>
>
> The privacy violations from the second step of the RLHF process has not been shown in the literature in the context of LLMs. However, we believe that linkage attacks can be carried out if one were not to make this step private.  This could be an interesting research direction in understanding the privacy leakage of LLMs.
>
>
> Finally, the last step of the RLHF process also involves making gradient updates to the model (using PPO-style algorithms), which in turn updates model weights. Moreover, we believe that one of the powers of PPO style algorithms and RLHF process in general is to continually improve the model behavior based on user interactions, which would involve training the models on user content and demonstrations. So, attacks similar to prior work (Duan et al., 2023; Carlini et al.,2021; 2019) should hold here.
>
>
> As per your suggestion, we will add these examples to make our motivation stronger.
>
>
> > Q: The experiments in the main body do not include error or number of trials details. It is unclear in Table 1 why models with less privacy should do worse than those with more privacy (GPT-2 Medium, eps 4->8, or GPT-2 Large eps 8->Inf). Such results demand further study and/or ablations and are difficult to interpret without confidence intervals.
>
>
> A: We appreciate your comments on the details regarding error margins and the number of trials in our experiments, as well as your remarks on the unexpected results in Table 1. Initially, due to computational constraints and the extensive nature of our experiments, we were only able to run each experiment once. In the time since our initial submission, we have dedicated further resources to conducting additional experiments, thereby enhancing the robustness of our findings in response to your valuable feedback. We have now rerun all experiments three times, each with different random seeds, to ensure the robustness of our results. The revised manuscript includes an updated Table 1 that now features 95% confidence intervals for each result.
>
>
> This additional experimentation has provided more clarity on certain previously ambiguous outcomes, such as those involving the GPT-2 Medium model with epsilon 4 outperforming epsilon 8. Our updated results demonstrate a consistent trend: models with less privacy generally exhibit better or at least similar performance, but not worse, compared to their more private counterparts. Additionally, we observed that larger models with less privacy display less variation in performance.
>
>
> An exception to this trend was noted with the non-private GPT-2 Large model. Despite extensive hyperparameter tuning, we were unable to achieve favorable non-private results for this specific model configuration. This anomaly could be an interesting area for future investigation, as it deviates from the general patterns observed in our study.
>
>
> > Q: The use of corporate imagery (Reddit / OpenAI) weakens the overall presentation and the generality of the results.
>
>
> A: We acknowledge the reviewer's concern regarding the use of specific corporate imagery and its potential effect on the perceived generality of our results. In light of your feedback, we have updated the images in the revised manuscript.

---

> > ### Comment · Reviewer_vWsq · 2023-11-21
> >
> > We appreciate the updates made to the submission and the clarifications provided.
> >
> > Note that the updated results are not properly reflected in the Contributions section in the main body. Also, it is unclear how the authors are able to compute a "95% confidence interval" with respect to their results. We would have expected these values to be in terms of either standard error or the variance of the recorded values.
> >
> > We maintain our score and look forward to the submission being rectified.

---

> > > ### Author Response · Authors · 2023-11-21
> > > **Official response to Reviewer vWsq**
> > >
> > > We thank the reviewer for bringing up the issue. We've updated the contributions section to properly reflect the updated results.
> > >
> > > For the 95% confidence interval -- to clarify, we first calculate the standard error (=standard deviation / sqrt(number_of_trials)), then multiply the standard error number by 1.96, which gives us the 95% confidence interval.
> > >
> > > Thanks again for your time and effort in reviewing our paper! We are happy to engage in further discussion if anything is still unclear.
> > >
> > > Best,
> > >
> > > Authors

---

> ### Author Response · Authors · 2023-11-21
> **Official response to reviewer vWsq (2/2)**
>
> > Q: Work in differential privacy and RL can be traced to differentially-private policy evaluation (Balle, Gomrockchi, Precup).
>
>
> A: Thank you for pointing out the relevant work in differential privacy and reinforcement learning, specifically the work “Differentially Private Policy Evaluation” by Balle, Gomrokchi, and Precup. We have included this reference in the related work section of our revised manuscript.
>
>
> > Q: The paper touches on the privacy accounting implications when T_PPO≠1, but does not offer to evaluate the implication of fixing it to the default value of 4. What are we losing by setting T_PPO≠1 instead of 4 in terms of utility? Is this trade-off significant?
>
>
> A: We appreciate the reviewer highlighting the importance of evaluating the implications of setting $T_{PPO} = 1$ in our DPPPO implementation. Indeed, setting $T_{PPO} = 1$ simplifies the privacy analysis and leverages privacy amplification by subsampling in the DPSGD algorithm. In response, we have conducted additional analyses and added the findings to Appendix B of our revised manuscript. Our investigation reveals that setting $T_{PPO} > 1$ does not result in a statistically significant improvement in utility. Therefore, based on our experimental findings, we conclude that the utility trade-off incurred by setting \( T_{PPO} = 1 \) is not substantial.
>
>
> We would be happy to engage in further discussions with the reviewer should any follow-up suggestions arise. We would also very much appreciate the reviewer considering increasing their rating in case they find our responses compelling.

---

### Official Review · Reviewer_A7MT · 2023-11-06

**Soundness:** 3 good
**Presentation:** 4 excellent
**Contribution:** 3 good
**Rating:** 8
**Confidence:** 3

**Summary:**

This paper offers an approach to align Large Language Models with human preferences and feedback via a privacy preserving RLHF methodology and perform some comparison experiments to show the correctness of their method.

**Strengths:**

(1) The approach proposed in this paper for aligning language models with PPO in a privacy-preserving way is original; (2) The paper is clearly written and well-organized. (3) The paper gives a quite comprehensive analysis of the procedure and emphasize the difficult issues in the implementation.

**Weaknesses:**

(1)The DP part is too condensed to understand.  The authors used DP-SGD on several occasions but without a clear explanation of this algorithm. And in the main text, I could not find a concrete DP algorithm and a clear procedure how it is combined with the alignment. (2) Algorithm 1 is not original. I don’t see the reason why it was presented in detail in the paper.  The PRIVATE alignment should be more interesting.

**Questions:**

(1)What is non-private REWARD model? What is the private REWARD model? DP mechanisms are not equivalent to adding noises.  Could you specify the DP mechanisms in the alignment?

---

> ### Author Response · Authors · 2023-11-21
> **Official response to reviewer A7MT**
>
> We thank the reviewer for their careful reading and thoughtful comments along with their positive score. We address all their questions below.
>
> > Q: The DP part is too condensed to understand. The authors used DPSGD on several occasions but without a clear explanation of this algorithm.
>
> A: This is a fair point, and we will elaborate about the DPSGD algorithm in the final version. We only gave a sketch of it in Section 2.2 due to page limit. As per your suggestion, we have now added more description of the DPSGD algorithm in Appendix A. We hope this helps.
>
> > Q:  And in the main text, I could not find a concrete DP algorithm and a clear procedure on how it is combined with the alignment.
>
> A: Thanks for your feedback. The alignment process involves 3 steps, and each of those steps need to be made DP, and finally one needs to do composition across these steps to ensure that the whole process is (epsilon, delta)-DP. We note that the concrete DP algorithm that we use is DPSGD in all 3 steps. We combine the PPO algorithm with DPSGD training and along the way make some algorithmic choices as described in Section 4 in the paragraph titled “DP Adaptation of PPO”. We would be happy to revise in further discussions with the reviewer should any follow-up suggestions arise.
>
> > Q:  Algorithm 1 is not original. I don’t see the reason why it was presented in detail in the paper.
>
> A: We agree that Algorithm 1 is not our contribution. We added this pseudocode in the main body due to two reasons: 1) PPO style algorithms are less familiar to DP audience, 2) We needed a reference to discuss the modifications and algorithmic choices we made that is described in the paragraph “DP Adaptation of PPO.”
>
> > Q: What is a non-private REWARD model? What is the private REWARD model?
>
> A: Non-private reward model and private reward model, which are both simply language models, refer to the way we train reward models, which is described in the II-step in Section 4. In the non-private reward model, we train the second step of the alignment process without DP guarantees (using SGD instead of DPSGD) whereas in the private reward model we train the reward model using DPSGD. For the whole framework to satisfy DP guarantees, we require that the second step also needs to be private as discussed in Section 4.
>
> We note that the non-private reward model is only used in our experiments to compare against efficacy of our DP framework.
>
> > Q: DP mechanisms are not equivalent to adding noises. Could you specify the DP mechanisms in the alignment?
>
> A: We are not entirely sure about what you mean by “DP mechanisms are not equivalent to adding noises”. If you can help us by elaborating more, we can answer your question better.
>
> Regarding DP mechanisms, all DP steps in our alignment process eventually boil down to making the SGD (or Adam) optimizer private, which we accomplish by adding Guassian noise. In this sense, all our mechanisms are based on composition of subsampled Gaussian mechanisms. While the first and second step in our alignment process is akin to training with DPSGD, the third step requires us to make some algorithmic choices. If we stick with the standard implementations of PPO algorithm, it would require us to change our privacy composition and analysis techniques, so, we changed the standard PPO implementation that helps us to have easier privacy analysis. This is described in Section 4 in the paragraph titled “DP Adaptation of PPO”.
>
> We would be happy to engage in further discussions with the reviewer should any follow-up suggestions arise.

---

### Comment · Area_Chair_QheC · 2023-11-21
**Engaging in discussion with the authors**

Dear reviewers, we are approaching the end of the discussion period (Nov 22) with the authors , please read the rebuttal and engage with authors to discuss any further questions/ clarifications you may have,

Many thanks

AC

---

### Meta-Review · Area_Chair_QheC · 2023-12-11

**Metareview:**

The paper combines DP-SGD with RLHF using PPO for a privacy preserving alignment. The paper advocates for a private learning of supervised fine-tuning and of  the reward model followed by  private learning in PPO. The paper uses LoRA in the alignment to update smaller numbers of parameters in the DP-SGD iterations.

In terms of contributions, the work is an application of previous works, and reviewers pointed out to differentially-private policy evaluation (Balle, Gomrockchi, Precup), that already combined RL with DP.

The paper is nicely executed in terms of combining these known ideas. The motivation of privacy in the alignment step was questioned by the reviewers, authors clarified that fine-tuning  LLMs from previous works (Duan et al., 2023; Carlini et al.,2021; 2019) can lead to memorizing user contents.

In response to reviewers feedback, authors added  1) uncertainty results by repeating experiments for three seeds,  2) impact of $T_{PPO} \neq 1$ on utility and added a Pareto front of utility versus privacy in the learning.

Reviewers questionned also the need of making PPO DP , if the reward model was already private, authors clarified that if the user prompts are considered private there is the need of making the PPO private.

In the context of private RLHF alignment this paper is a good contribution but not a significant one:  the chaining of differential privacy from supervised fine-tuning to reward model and then in PPO , will not result in best privacy/ utility tradeoffs. Direct Policy optimization, circumvents the learning of a reward model and a straightforward DP-SGD on DPO is expected to outperform the method presented in this paper.

**Justification For Why Not Higher Score:**

The contribution of the paper is incremental but it is a well executed study.

**Justification For Why Not Lower Score:**

The contribution of the paper is incremental but it is a well executed study.

---

### Decision · Program_Chairs · 2024-01-16

Accept (poster)